# Controlled spin switching in a metallocene molecular junction

M. Ormaza [1], P. Abufager[2], B. Verlhac[1], N. Bachellier[1], M.-L. Bocquet [3], N. Lorente[4,5] & L. Limot[1]

The active control of a molecular spin represents one of the main challenges in molecular spintronics. Up to now spin manipulation has been achieved through the modification of the molecular structure either by chemical doping or by external stimuli. However, the spin of a molecule adsorbed on a surface depends primarily on the interaction between its localized orbitals and the electronic states of the substrate. Here we change the effective spin of a single molecule by modifying the molecule/metal interface in a controlled way using a low-temperature scanning tunneling microscope. A nickelocene molecule reversibly switches from a spin 1 to 1/2 when varying the electrode–electrode distance from tunnel to contact regime. This switching is experimentally evidenced by inelastic and elastic spin-flip mechanisms observed in reproducible conductance measurements and understood using first principle calculations. Our work demonstrates the active control over the spin state of single molecule devices through interface manipulation.

[1] Université de Strasbourg, CNRS, IPCMS, UMR 7504, F-67000 Strasbourg, France. [2] Instituto de Física de Rosario, Consejo Nacional de Investigaciones Científicas y Técnicas (CONICET), Universidad Nacional de Rosario, Bv. 27 de Febrero 210bis S2000EZP, Rosario, Argentina. [3] PASTEUR, Département de Chimie, Ecole Normale Supérieure, UPMC Univ. Paris 06, CNRS, PSL Research University, Sorbonne Universités, 75005 Paris, France. [4] Centro de Fsíca de Materiales CFM/MPC (CSIC-UPV/EHU), Paseo Manuel de Lardizabal 5, 20018 Donostia−San Sebastián, Spain. [5] Donostia International Physics Center (DIPC), Paseo Manuel de Lardizabal 4, 20018 Donostia−San Sebastián, Spain. Correspondence and requests for materials should be addressed to M.O. (email: ormaza@ipcms.unistra.fr) or to L.L. (email: limot@ipcms.unistra.fr)

The ability to modify the spin of a molecule in a controlled and reversible way is essential to develop novel spin-based technologies[1]. Although the spin of surface-supported single molecules is mainly determined by molecule/surface interactions[2–5], it can be modified by external parameters such as the mechanical[6] or the chemical modification of the molecule[7–12], the application of electric fields[13–15], light or temperature[16,17]. The effect of approaching the metallic tip of a scanning tunneling microscope (STM) to the molecule is a way to alter the local environment of the molecule[18,19]. It can result in structural relaxations[20], or even modify the molecular configuration through the application of bias pulses[21], ultimately changing the spin state of the molecule. However, determining the factors leading the spin modification of a molecule when adsorbed on a surface or when coupled between metallic electrodes is a complex process in which different subtle contributions need to be considered.

Scanning tunneling spectroscopy (STS) is a useful tool to determine the spin of surface-supported magnetic single objects by exploring the zero-bias anomalies produced by different spin-flip scattering mechanisms[22,23]. While the Kondo screening involves the elastic spin-flip scattering of conduction electrons with the magnetic impurity below a critical temperature $T_K$[24], the magnetic anisotropy of single objects leads to an inelastic scattering of these electrons. In fact, the magnetic anisotropy of surface-supported objects—atoms or molecules—has been attracting a growing interest in relation to ultra-dense storage technology[25] and quantum computing[26]. From a practical viewpoint, a uniaxial magnetic anisotropy energy of the form $DS_z^2$ provides stable magnetic states into which information can be encoded and may, moreover, be externally controlled[11,21,27–34]. The general picture to address is that single objects have the potential for both magnetic anisotropy and the Kondo effect[35], the outcome of their interplay depending on the object's spin[23,36–38] and on the relative weight of $k_B T_K$ vs. $D$[39,40].

Elastic and inelastic tunneling spectra can be used to probe the above mentioned interplay in an environment that can be characterized with atomic-scale precision. In recent experiments[11,12,31–33], different surface topologies were used to tune the Kondo exchange interaction between the spin and the underlying

metal, while spin control was possible through hydrogen doping of transition metal atoms.

Here, we control the spin of a single-metallocene molecule in a junction using a STM. We attach a $S = 1$ molecule with an easy-plane magnetic anisotropy ($D > 0$), nickelocene ($Ni(C_5H_5)_2$, noted Nc hereafter), to the tip apex of a STM and form an atomically precise contact with a Cu(100) surface. The molecular spin switches from 1 to 1/2 when we go from the tunnel to the contact transport regime, as monitored by the two-order of magnitude change in the differential conductance near the Fermi level. This abrupt difference comes from the different nature of the spin–flip scattering that governs the conductance around the Fermi energy of the junction in the two regimes. Our findings are corroborated by density functional theory (DFT) calculations.

## Results

**Tunnel regime.** An image of the Cu(100) surface after a molecular deposition is shown in Fig. 1a, in which isolated Nc decorate the surface terraces. The molecules show a ring-shaped pattern, indicating that one cyclopentadienyl ring ($C_5H_5$, noted Cp hereafter) is exposed to vacuum, i.e., the long molecular axis of Nc is perpendicular to the Cu(100) surface[41].

We found that in order to contact nickelocene between both electrodes, surface and tip, it is more stable to first transfer the molecule from the metallic surface to the tip, by approaching the tip to the target molecule at −1 mV and 50 pA[34]. Prior to the molecular transfer, care was taken to select a monoatomically sharp tip apex[42]. Information may be gathered on the status of the Nc at the tip apex by acquiring counter-images[43], which consist in imaging single atoms on the surface with the molecular tip. The typical molecular pattern observed in the counter images of Cu adatoms is presented in Fig. 1b and differs from the featureless protrusion observed with a metallic tip (encircled in Fig. 1a). The pattern reveals that the tip is terminated by a Cp ring of a tilted Nc molecule, the tilt angle exhibiting some tip dependency. To confirm this assignment we mimicked the molecular tip through DFT calculations by considering a Nc molecule adsorbed on a Cu atom on Cu(100) (inset of Fig. 1c, see Methods). The molecule is undeformed, tilted by 13° with respect

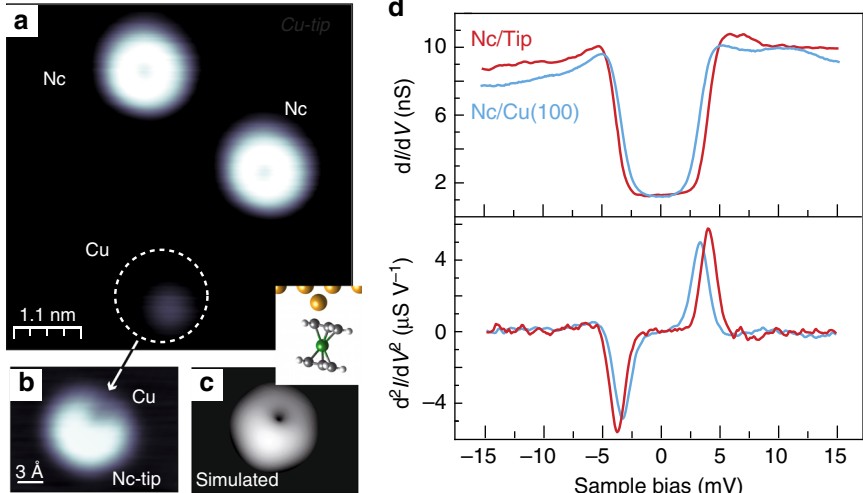

**Fig. 1** Nickelocene adsorption on Cu(100) and on the tip apex. **a** STM image acquired with a metallic tip (5.6 × 5.6 nm², −15 mV, 20 pA). **b** Counter image of the Cu atom obtained with a Nc-terminated tip (1.7 × 1.3 nm², −15 mV, 20 pA). **c** Calculated relaxed configuration of a Nc molecule on top of a Cu adatom on Cu(100), together with the simulated STM image. Atom colors: Cu (yellow), C (gray), H (white) and Ni (green). **d** d*I*/d*V* spectra and their derivative acquired with a Cu-tip over a Nc molecule (solid blue line) and with a Nc-terminated tip over the Cu(100) surface (solid red line). Feedback loop opened at 100 pA and −15 mV

to the surface normal and linked through two C atoms to the Cu atom. The corresponding simulated image in Fig. 1c is in good agreement with the experimental counter-image of Fig. 1b. Note that in a previous work[34] in order to determine the magnetic anisotropy of a Nc on top of a Cu atom, the molecule was forced to be centered with respect to the Cu adatom underneath. However, the current results show that the relaxed situation is the one in which the molecule is tilted.

The top panel of Fig. 1d presents the $dI/dV$ spectrum acquired with a Nc-tip above Cu(100). For comparison, we have included the $dI/dV$ spectrum acquired with a metallic tip above a Nc molecule on Cu(100). Both spectra were acquired with a lock-in amplifier using a frequency of 716 Hz and an amplitude of 150 $\mu$V rms; the metallic tip was verified to have a flat electronic structure in the bias range presented. The steps, symmetric with respect to zero bias, that are visible in the $dI/dV$ correspond to the manifestation of efficient spin–flip excitations within the molecule[34]. These occur between the ground state $|S = 1, M = 0\rangle$ and the doubly degenerate $|S = 1, M = \pm 1\rangle$ excited states of the molecule, the threshold energy observed corresponding to the longitudinal magnetic anisotropy energy, $D$. The bottom panel shows the numerical derivative of the $dI/dV$ spectra in order to facilitate the identification of the threshold values. For Nc on the surface we find $D = (3.2 \pm 0.1)$ meV, while for Nc on the tip the magnetic anisotropy grows to $D = (3.7 \pm 0.3)$ meV. This reflects changes in the local ligand field induced by the different adsorption configuration[21,30,34].

**Contact regime**. We then engineer a molecular junction by bringing the Nc-tip into contact with Cu(100). For an increased control over the molecular junction, the contact is always performed atop a copper surface atom thanks to the use of atomically-resolved images (Inset of Fig. 2b). These images are routinely acquired by scanning the molecular tip while in contact with the surface[44,45], which can be taken as an indication of the robustness of a Nc-tip to external strain. The right panel of Fig. 2a presents the conductance ($G$) vs. tip displacement ($z$) curve acquired at a fixed bias of −2 mV with a Nc-tip vertically displaced towards the surface. For the $G(z)$ measured at −2 mV, an abrupt increase of $G$ by more than a factor 10 ($G = 0.7$) reveals the transition, indicated by a dashed line, between the tunneling and the contact regime (see Supplementary Fig. 1). As we show below, the sudden change in $G$ is exclusively driven by a spin switch of nickelocene. Note that here the tip is moved from its initial tunneling position $z = -2.1$ Å ($G = 2 \times 10^{-4}$ in units $2e^2h^{-1}$) up to the contact point $z = 0$ ($G = 0.04$). Once the contact is established, if we reverse the process and increase the tip–sample distance, we do not find the contact point exactly at the same $z$. Typical variations of the contact point are around 10–20 pm, indicating some hysteresis effects. Notice also that the exact values of the conductance as well as of the contact position under the same conditions present some tip dependency.

To elucidate this finding, we present in the left panel of Fig. 2a a two-dimensional intensity plot of a series of $dI/dV$ spectra acquired at varying $z$. For completeness, Fig. 2b presents the spectra for decreasing tip–surface distances. The spectra reveal the presence of spin-flip scattering events, whose nature may be reversibly controlled via $z$. Within the tunneling regime, the spectra exhibit inelastic spin-flip excitations as anticipated in Fig. 1d. We note an enhancement of the differential conductance at voltages corresponding to the excitation threshold, which is increasingly pronounced as the contact point is approached ($z >$ −0.5 Å). Recent studies have pointed out that this enhancement could reflect many-body effects including Kondo-like phenomena[23,37,38]. However, this scenario also requires a sizable

reduction of the magnetic anisotropy energy, and therefore of the excitation threshold[11,31], at variance with our experimental observations. The enhancement here rather reflects a stationary population of excited states[23,46–48] due to the increasingly large currents flowing through the junction when approaching contact. This so-called spin pumping can be expected to be significant due to the efficiency of the spin-flip excitations in Nc produced by tunneling electrons[34], or more generally, in molecules[49–51].

The most striking result is found once the molecular contact to the surface is established ($z > 0$). As we anticipated above, a sudden increase in the conductance occurs at zero bias (Fig. 2a), which we ascribe to the presence of a Kondo resonance. As shown in Fig. 2a, a sharp peak appears in the $dI/dV$, while the inelastic excitation thresholds are lost. The emergence of a spin-1 Kondo effect in the presence or in the absence of positive magnetic anisotropy is in principle possible[39,40]—the former case being unlikely here as the resonance does not split apart[29]. However, we find that the system actually behaves as a spin 1/2-system. The peak is perfectly fitted by a Frota function (see Supplementary Fig. 2 and Supplementary Table 1), which is close to the exact line shape expected for a spin-1/2 Kondo system[52]. From the fit, we find that the peak is centered at $\epsilon_K = (0.0 \pm 0.1)$ meV for all used tips up to the highest tip excursion explored (0.8 Å). The line width, therefore $T_K$, increases nearly exponentially with $z$ (Fig. 2c), similarly to other junctions comprising a single Kondo impurity[53–55]. Even though one would expect a linear increase of the $d$-level broadening with $z$ considering the behavior of the Kondo temperature, here we have seen that the molecular levels follow a complex behavior due to the involved interaction with the tip (see Supplementary Fig. 3). To further confirm the spin-1/2 nature of the Kondo effect, we recall that the resonance amplitude (noted $\sigma$) should be a universal function of the normalized temperature $T/T_K$. For a quantitative analysis we therefore fit the curve in Fig. 2d to the function $\sigma = [1 + (T/T_K)^2 (2^{1/s} - 1)]^{-s}$ (in units of $2e^2h^{-1}$)[56] and extract $s = (0.29 \pm 0.02)$, in remarkable agreement with the spin-1/2 Kondo effect of semiconductor quantum dots[57]. The amplitude of the resonance is close to the unitary limit and indicates a complete Kondo screening. Note that in a previous work[55] non-equilibrium effects were shown to be absent in the Kondo resonance even at contact. Our findings therefore strongly support the emergence of a spin-1/2 Kondo effect where there is no magnetic anisotropy. To elucidate its origin, here below, we show through DFT calculations that Nc switches its spin from 1 to 1/2 upon contact with the surface.

**Density functional theory calculations**. The DFT calculations were performed by modeling the molecular tip by a Nc atop a Cu atom adsorbed on a Cu(100) plane (Fig. 3a). The molecular tip was placed at different distances from a Cu(100) surface and the junction was fully relaxed. Figure 3c–e shows the resulting molecular junctions for the three most representative configurations (see frontal views in Supplementary Fig. 4). In Fig. 3c, the distance between the two Cu atomic planes (noted $d$) is 12.17 Å and corresponds to the tunneling regime (see Fig. 1c) described previously. Figure 3d shows the molecular junction at $d = 11.14$ Å and corresponds to the transition between the tunneling and contact regimes. The transition point has been assigned to the point in which a change in the behavior of structural parameters is observed, and has been confirmed by the theoretical transmission probabilities (Fig. 3f). The molecule is distorted at the transition, but still bonded to the Cu atom through two C atoms. Finally, Fig. 3e corresponds to $d = 9.66$ Å and is representative of the molecular junction in the contact regime. The molecule exhibits Cp rings that are parallel to both metallic planes and the Cu atom is coordinated to five C atoms. Figure 3b quantifies the

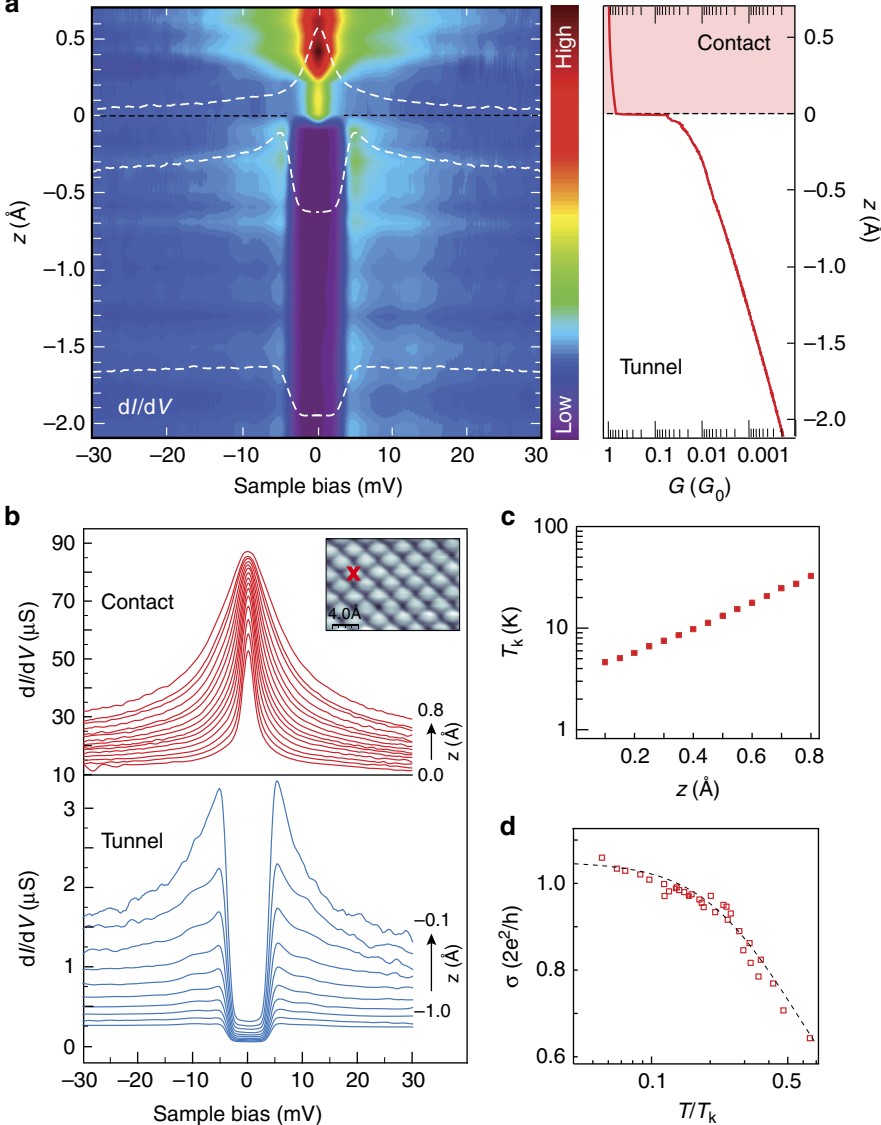

**Fig. 2** Spectroscopy of the molecular junction. **a** Left panel: 2D intensity plot of d$I$/d$V$ spectra acquired with a Nc-tip at increasing $z$. The intensity of the spectra has been normalized by the opening conductance value (at −30 mV) of the spectrum at $z = 0$ Å. As a guideline to identify the differences, three characteristic spectra (white dashed lines) are superimposed. Right panel: $G$-vs.−$z$ curve measured at −2 mV for a Nc-tip approaching the Cu(100) surface. The boundary between the tunnel and the contact regime occurs at $z = 0$ pm and is indicated with a dashed line. **b** Individual d$I$/d$V$ spectra in the tunnel (bottom panel) and contact (top panel) regimes for several $z$. Note that right after contact, the width of the Kondo resonance is smaller than the inelastic excitation threshold. Inset: Contact image of the Cu(100) surface using a Nc-terminated tip ($2 \times 1.4$ nm$^2$; 300 pA, 30 mV). The cross indicates the contact point. **c** Evolution of $T_K$ with $z$. The Kondo temperature is 4.5 K or lower at $z = 0$, and 32.7 K at $z = 0.8$ Å. **d** Evolution of the resonance maximum $\sigma$ with $T/T_K$, where $T = 2.4$ K is the working temperature. $\sigma$ and $T_K$ are extracted from the Frota fits (see Supplementary Fig. 2)

structural changes with $d$. As shown, at the transition point the distances between the Cu and Ni atom with respect to the tip electrode, noted $d_1$ and $d_1 + d_2$ respectively, are maximum, as well as the tilt difference between the Cp rings, noted $\Delta\Theta$. The distance between the Ni atom and the surface-electrode (noted $d_3$) instead decreases continuously with $d$. Experimentally, we have seen that in the tunneling regime, despite the slightly different initial orientations that the molecule might have on the tip, similar results are obtained in the d$I$/d$V$ spectra for different molecular tips when going to contact. This indicates that the Nc molecule tends to adopt always the same configuration, with parallel Cp rings, when contacted between the tip and the surface. From Fig. 3f we can obtain the ratio between the total transmission probabilities for the different spin channels (spin-up $T_\uparrow$, spin-down $T_\downarrow$) and find a −74% spin polarization in the tunneling regime and −55% in contact, where the spin polarization is $\frac{T_\uparrow - T_\downarrow}{T_\downarrow + T_\uparrow}$. The decreased value at contact is in agreement with the trend found in a previous theoretical study[58].

Figure 3g presents how the magnetization of the Nc molecule, the Ni atom and the Cu atom change with $d$. In the tunneling regime, the calculated magnetic moment for Nc is 1.7 $\mu_B$, the Ni atom carrying 1 $\mu_B$. At the transition, these values decrease to 1 and 0.7 $\mu_B$ respectively, and afterwards, at contact, stabilize around 0.75 $\mu_B$ for Nc and 0.5 $\mu_B$ for the Ni atom. The Cu atom remains non-magnetic during the contact process. The initial magnetic moment of the molecule is halved when contacted between the two electrodes, meaning that the molecular spin changes from $S = 1$ to $S = 1/2$. Such a spin switch is in agreement with the spectroscopic fingerprints highlighted experimentally.

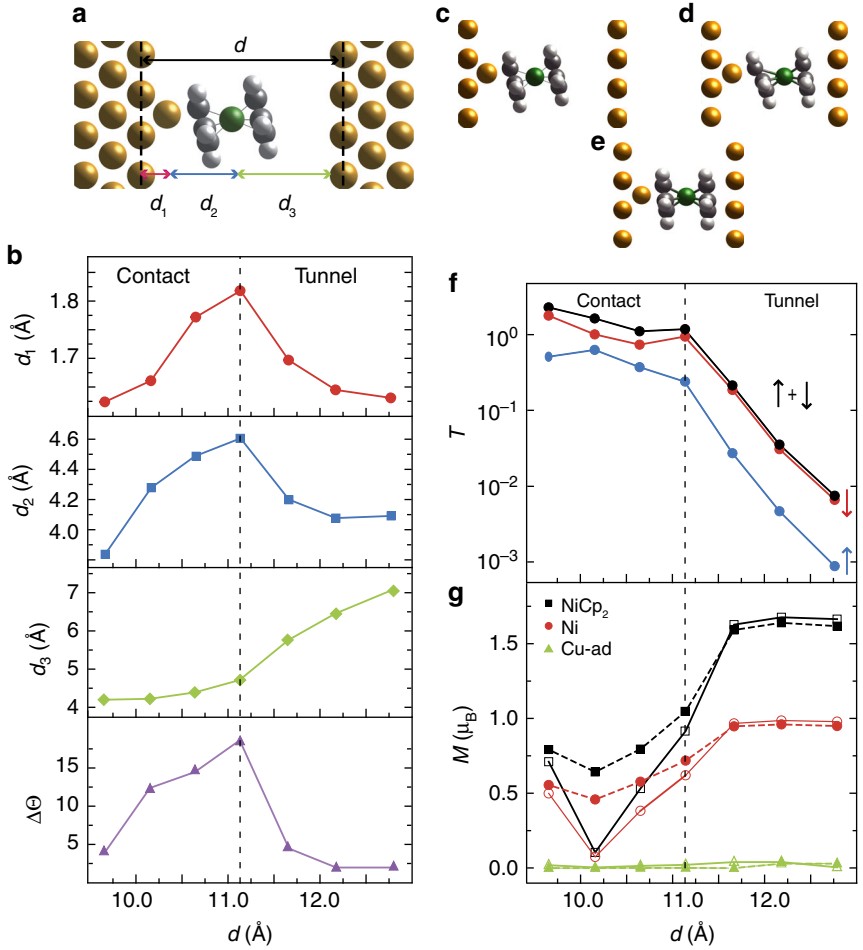

**Fig. 3** Structural and magnetization analysis of the molecular junction. **a** Structure of the calculated molecular junction, including the main relevant distance parameters. **b** The panels show how the structural parameters $d_1$, $d_2$, $d_3$, $\Delta\Theta$ change as a function of $d$. The structure of the molecular junction for: **c** the tunneling regime ($d = 12.17$ Å), **d** the transition between the tunneling and the contact regimes ($d = 11.14$ Å), and **e** the contact regime ($d = 9.66$ Å). **f** Total transmission (black line) and the transmission per spin channel (red and blue lines) at the Fermi level for the explored configurations. **g** Magnetization of: Nc (black), Ni (red), Cu atom (green) as a function of $d$. Full symbols correspond to SIESTA (Mulliken) results and open symbols to VASP (Bader) results

The change on the molecular charge with respect to the gas phase from tunneling ($-0.12e^-$) to contact ($+0.07e^-$) is not enough to explain such a change on the magnetization. To scrutinize the effect of the deformation of the molecule on the molecular magnetization, the magnetization of the isolated Nc molecule was computed using the geometrical configurations found previously in the molecular junction when changing $z$ (Fig. 3c–e). No relevant difference was observed for the different configurations, indicating that the magnetization change is neither driven by the molecule-substrate charge transfer nor by the deformation suffered by the molecule.

Figure 4a presents the spin-resolved transmission for the tunnel (thick line) and contact (thin line) regimes. The transmission of Fig. 4a implies that the transport is mainly due to the hybridization of surface electronic states with the frontier molecular orbitals, which are $d_{xz}$—and $d_{yz}$—based molecular orbitals (see Supplementary Fig. 5)[34]. To get a deeper understanding of the different features observed in the transmission function, the density of states projected (PDOS) onto the C(2p) and Ni(3d) atomic orbitals are shown in Fig. 4b. The clear correspondence between spin up and spin down peaks on the transmission function and the PDOS (Fig. 4b) makes it possible to assign both peaks to the transmission through the spin-polarized degenerate frontier orbitals. The comparison between contact and tunnel results shows that the peaks associated to the frontier-orbital in both spin up and spin down channels shift towards the Fermi level in the contact regime, reducing the exchange splitting energy due to the screening of the intra-orbital Coulomb repulsion ($U$). This is due to the enhancement of the molecule–electrode interaction that also induces a more pronounced broadening of the molecular levels. Such increase of the hybridization ($\Gamma$) reduces the charge in the majority spin and increases the minority spin occupation. Our analysis then shows that both effects, namely, the increase of $\Gamma$ and the reduction of $U$ contribute to halve the magnetic moment of the molecule in the contact regime. Therefore, the spin transition can be explained by the coupling to the substrate of the frontier orbitals located close to the Fermi level.

## Discussion

In summary, we have shown how the spin of a Nc molecule and, associated, spin-flip scattering from Nc can be reversibly modified via a controlled contact to a copper electrode. The spin transition takes place due to the enhanced electronic screening following the Nc-surface contact. On a more general note, the portability of the Nc molecule or eventually of other metallocenes[59] via the STM tip offers novel opportunities for testing how surface-supported objects modify the molecular magnetism of these molecules.

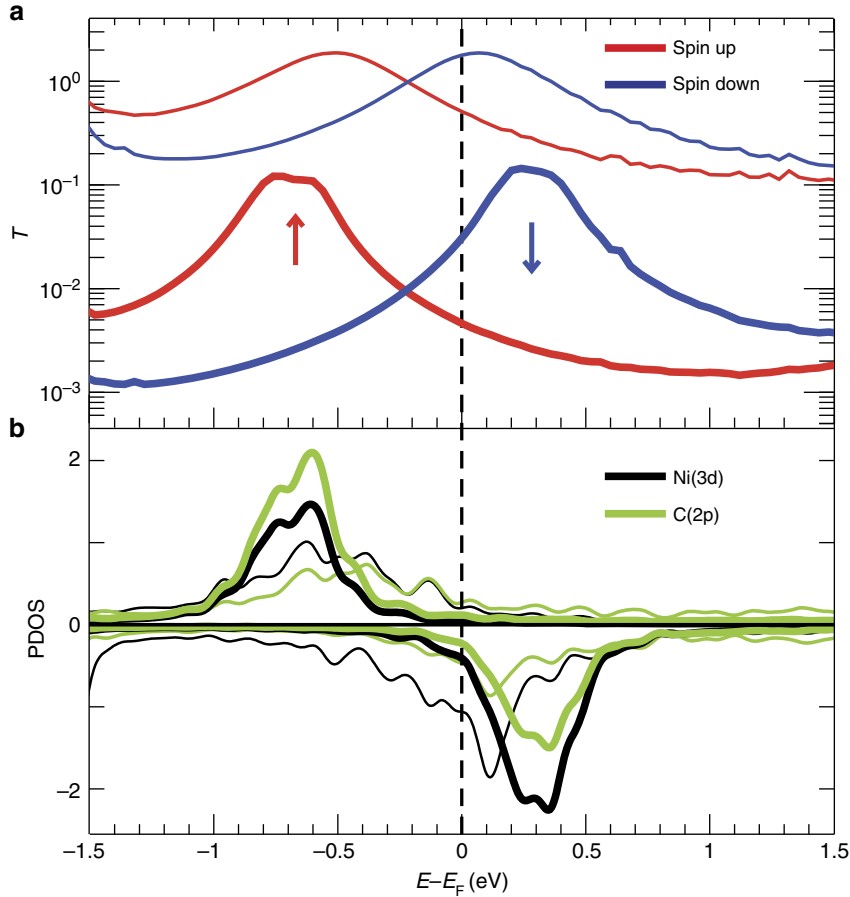

**Fig. 4** Transmission and PDOS of the molecular junction. **a** Spin-resolved electron transmission as a function of electron energy with respect to the Fermi energy for the Nc-tip above Cu(100). The thick line corresponds to the tunneling configuration (Fig. 3c) and the thin line to the contact configuration (Fig. 3d). **b** Density of states projected (PDOS) onto the C(2p) and Ni(3d) atomic orbitals for the tunneling (thick line) and for the contact (thin line) regimes. A comparison between SIESTA and VASP results can be found in Supplementary Fig. 6

## Methods

**STM measurements**. The experiments were performed in an ultra-high vacuum STM operated at 2.5 K. A clean Cu(100) surface was prepared by repeated cycles of Ar⁺ sputtering and annealing up to 800 K at a base pressure of $2 \times 10^{-10}$ mbar. Nickelocene molecules were deposited from a crucible onto the cold copper surface (below 80 K) at a rate of 0.025 ML/min resulting in a sub-monolayer coverage. Directly after molecular deposition, the sample was inserted in the STM. Analysis of STM data were performed with the WSxM software[60].

**Calculation details**. Calculations were carried out using two DFT packages. VASP[61–66] has been used to explore the adsorption of the Nc molecule on a STM tip modeled by a Cu(100) surface with a Cu adatom acting as a monoatomic tip apex. Geometrical effects when a second electrode (a Cu(100) surface) was approached have been evaluated also with VASP (see Fig. 3a). SIESTA[67] calculations confirmed VASP results about the electronic and magnetic properties of the Nc-terminated tip/Cu(100) interface and allowed us to perform electronic transport calculations using TRANSIESTA[68].

We optimized the structure of the Nc-terminated tip/Cu(100) interface using DFT at the spin-polarized generalized gradient approximation (GGA-PBE) level, as implemented in VASP[61–66]. In order to introduce long-range dispersion corrections, we employed the so-called DFT-D2 approach proposed by Grimme[69]. We used a plane wave basis set and the projected augmented wave (PAW) method with an energy cutoff of 400 eV. The two surfaces representing substrate and tip were modeled using a slab geometry with a 4 × 4 surface unit cell and 6 layers for the surface holding the tip–apex and the molecule, and 7 layers for the approaching surface electrode.

The valence–electron wavefunctions were expanded in a basis set of local orbitals in SIESTA. A double-$\zeta$ plus polarization (DZP) basis set was used to describe the Nc molecule and surface-atom electrons. Diffuse orbitals were used to improve the surface electronic description and a single-$\zeta$ plus polarization basis set for the copper electrodes. The use of a DZP basis set to describe the adsorbate states is mandatory in order to yield correct transmission functions[70]. The k-point sampling was converged at 3 × 3, although the sampling was 11 × 11 for the

transmission calculations. These transport calculations were carried out from first-principles with a method based on non-equilibrium Green's functions combined with DFT as implemented in the TranSIESTA package[68]. The Ni pseudopotential used in the SIESTA calculations was extracted from reference[71].

**Data availability**. The authors declare that relevant data supporting the findings of this study are available on request.

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

## Acknowledgements

This work has been supported by the Agence Nationale de la Recherche (Grant Nos. ANR-13-BS10-0016, ANR-11-LABX-0058 NIE, ANR-10-LABX-0026 CSC) the ANPCyT project PICT Bicentenario No. 1962, the CONICET project PIP 0667, and the UNR project PID ING235. We acknowledge computer time provided by the CCT-Rosario Computational Center and the Computational Simulation Center (CSC) for Technological Applications, members of the High Performance Computing National System (SNCAD, MincyT-Argentina). Financial support from MINECO MAT2015-66888-C3-2-R and FEDER funds is also gratefully acknowledged.

## Author contributions

M.O., B.V., N.B. and L.L. carried out the experiments. M.O. and L.L. analyzed the experiments. P.A., N.L. and M.-L.B. performed the DFT calculations. M.O. and L.L. wrote the manuscript with comments of all authors.

## Additional information

**Competing interests:** The authors declare no competing financial interests.

