## [Peer Review File · Nature Communications]

Reviewers' comments:

Reviewer #1 (Remarks to the Author):

The manuscript "Controlled spin switching in a metallocene molecular junction" by M. Ormaza and co-workers is a nice low-temperature scanning tunneling microscopy and spectroscopy study in which the authors show how to change the effective spin of a single metallorganic molecule attached to the apex of the tip by approaching the sample surface. During tunneling conditions the effective spin is $S=1$ as evidenced by inelastic spin-flip spectroscopy while it changes to $S=1/2$ in the contact regime where a many particle Kondo resonance emerges at Fermi energy.

The manuscript is nicely written and the figures are clear and coherent. Therefore I believe that it will be of interest for the broad readership of Nature Communications. Prior publications, however, I would recommend the authors to consider to address my critiques and comments outlined below:

(1) On page 2, line 49, the authors write: "We attach a $S = 1$ molecule with an easy-axis magnetic anisotropy...". This is a typo; the molecule contains an easy-plane magnetic anisotropy as the authors also write in their previous publication [36].

(2) Figure 1(a) and (b) uses a very unconventional color-scale. Either the authors should change it to gray-scale as in panel (c), or, alternatively, put the color-scalebar into the figure.

(3) The authors write that they can reversibly switch the spin-state via the tip-sample distance z . Have the authors observed hysteresis effects? Or is the jump from one state to the other for a given junction independent of the sweep direction always at the same z distance? The authors should state either behavior more clearly in the text.

(4) Figure 2(a), right panel, shows $G(z)$ at -2mV , i.e. at a bias where the conductance is in particular low for the $S=1$ because it is smaller than the spin-flip excitation energy, and in particular large for the $S=1/2$ because it is inside the resonance of the Kondo peak.

I would find it very enlightened to also see such a $G(z)$ curve obtained at higher bias, above the threshold of the excitation energy and the resonance peak.

(5) Fig. 2(c) shows an exponential increase of the Kondo temperature with z . This suggest a linear increase of the d-level broadening with z . I'm wondering now, if the DFT calculation supports this claim. If so, I would highly recommend to include this into the paper.

(6) The authors address the change of the dI/dV spectra in the $S=1$ regime (Fig.2b bottom) to "spin pumping" (page 3, line 103). Have the authors tried to estimate from this data the lifetime of the excited state in a manner similar as shown by Loth et al. (ref. 49)?

(7) There is a recent paper by Jacobson et al. (Science Advances 3, e1602060 (2017)) which also report on switching the spin state from $S=1$ to $S=1/2$ by the amount of hydrogen atoms on a Co atom. Even though that the switching is due to a different effect, the authors should consider citing this work in their introduction.

Reviewer #2 (Remarks to the Author):

This manuscript reports a very interesting combined scanning tunnelling spectroscopy and density functional theory study of the active control of the spin moment of an adsorbed molecule through

approach by a tip of a scanning tunnelling microscope. The spin moment is investigated by the signatures of inelastic and elastic spin-flips in the tunnelling current and from density functional theory calculations. They find a spin-switch from a spin of 1 to $\frac{1}{2}$ from the tunnelling to the contact regime which is explained by the change in hybridization between the frontier orbitals and the substrate states. I find that their study is scientifically sound and that their conclusions are well-supported. Their finding is new and an important advance, and should be of interest for researchers in spintronics and nano-magnetism. The manuscript can be published after the authors have considered my minor points below.

In Fig. 2(a) there is no colour coding of the 2D intensity plot.

In the main manuscript it is not clear how the d_{xz} and d_{yz} orbitals are oriented with respect to the tip-substrate geometry.

There are spelling mistakes in the manuscript that need to be corrected.

Comments of Reviewer 1

Reviewer 1 finds the results presented in the paper of interest for the broad readership of Nature Communications. He/She recommends before publication to consider the following critiques and comments that we address next:

Comment 1:

On page 2, line 49, the authors write: “We attach a $S = 1$ molecule with an easy-axis magnetic anisotropy...”. This is a typo; the molecule contains an easy-plane magnetic anisotropy as the authors also write in their previous publication [36].

Author Reply 1:

We thank the referee for pointing out this typo. The molecule has an easy-plane magnetic anisotropy, this is, it has an easy plane perpendicular to the symmetry axis z . We have corrected the typo in the text.

Comment 2:

Figure 1(a) and (b) uses a very unconventional color-scale. Either the authors should change it to gray-scale as in panel (c), or, alternatively, put the color-scale bar into the figure.

Author Reply 2:

As recommended by the referee, we have changed the STM figures to a more conventional grey-scale.

Comment 3:

The authors write that they can reversibly switch the spin-state via the tip-sample distance z . Have the authors observed hysteresis effects? Or is the jump from one state to the other for a given junction independent of the sweep direction always at the same z distance? The authors should state either behavior more clearly in the text.

Author Reply 3:

We thank the reviewer for this remark. Depending on the specific tip we have, different z contact values are obtained. Nevertheless the z contact values differ at most by 50 pm under the same opening conditions. Additionally there are hysteresis effects. For a given tip, the z contact value is not the same depending if the tip is moved from tunnel to contact or the other way around. The hysteresis is small, typically of 20 pm. We have included this experimental remark in the main text.

We have changed the text (line 92 –line 101) as follows:

The right panel of Fig. 2 (a) presents the conductance (G) versus tip displacement (z) curve acquired at a fixed bias of -2mV with a Nc-tip vertically displaced towards the surface. For the $G(z)$ measured at -2mV , an abrupt increase of G by more than a factor 10 ($G = 0.7$) reveals the transition, indicated by a dashed

line, between the tunneling and the contact regime (see Supplementary Note 0). As we show below, the sudden change in G is exclusively driven by a spin switch of nickelocene. Note that here the tip is moved from its initial tunneling position $z = -2.1 \text{ \AA}$ ($G = 2 \times 10^{-4}$ in units $2e^2/h$) up to the contact point $z = 0$ ($G = 0.04$). Once the contact is established, if we reverse the process and increase the tip-sample distance, we do not find the contact point exactly at the same z . Typical variations of the contact point are around 10-20 pm, indicating some hysteresis effects. Notice also that the exact values of the conductance as well as of the contact position under the same conditions present some tip dependency.

Comment 4:

Figure 2(a), right panel, shows $G(z)$ at -2mV, i.e. at a bias where the conductance is in particular low for the $S=1$ because it is smaller than the spin-flip excitation energy, and in particular large for the $S=1/2$ because it is inside the resonance of the Kondo peak.

I would find it very enlightening to also see such a $G(z)$ curve obtained at higher bias, above the threshold of the excitation energy and the resonance peak.

Author Reply 4:

We present now the -30 mV curve as supplementary information (Supplementary Note 0). As we show, at -30 mV there is no abrupt change in conductance as for -2 mV. This is somewhat expected since when the bias is increased beyond the excitation threshold, the inelastic channel starts contributing to the conductance. Its contribution exceeds by a factor 5 the one from the elastic channel.

Comment 5:

Fig. 2(c) shows an exponential increase of the Kondo temperature with z . This suggests a linear increase of the d-level broadening with z . I'm wondering now, if the DFT calculation supports this claim. If so, I would highly recommend to include this into the paper.

Author Reply 5:

We agree with the referee that a single level would show a linearly increasing width with distance for an exponentially increasing Kondo temperature. However, here we have two levels that break their degeneracy (at least partially) and behave in a rather complex way (see Fig.S4, SI). The levels shift and broaden in a non-monotonous way, for example: the peak energy (ϵ) and width (Γ) are $\epsilon = -0.5 \text{ eV} \pm 0.1 \text{ eV}$ and $\Gamma = 0.25 \text{ eV} \pm 0.1 \text{ eV}$ for a tip-surface distance of 9.66 \AA , this increases to $\epsilon = -0.4 \text{ eV} \pm 0.1 \text{ eV}$ and $\Gamma = 0.4 \text{ eV} \pm 0.1 \text{ eV}$ at 10.16 \AA , but drops back to $\epsilon = -0.5 \text{ eV} \pm 0.1 \text{ eV}$ and $\Gamma = 0.35 \text{ eV} \pm 0.1 \text{ eV}$ for 10.65 \AA . Following the indication of the referee, we have added a small comment on the paper.

On page 5 (lines 123-125), we have added to the text the following sentence (marked in blue here below):

The line width, therefore T_K , increases nearly exponentially with z [Fig. 2(c)], similarly to other junctions comprising a single Kondo impurity [45–47]. Even though one would expect a linear increase of the d-level broadening with z considering the behavior of the Kondo temperature, here we have seen that the molecular levels follow a complex behavior due to the involved interaction with the tip (see Supplementary Note 2). To further confirm the spin-1/2 nature of the Kondo effect, we recall that the resonance amplitude (noted σ) should be a universal function of the normalized temperature T/T_K .

Comment 6:

The authors address the change of the dI/dV spectra in the $S=1$ regime (Fig.2b bottom) to “spin pumping” (page 3, line 103). Have the authors tried to estimate from this data the lifetime of the excited state in a manner similar as shown by Loth et al. (ref. 49)?

Author Reply 6:

Following the referee’s recommendation, we have estimated the lifetime of the excited state by fitting the experimental data at various tip excursions z by using the analytical approach of Loth et al. (see lifetime.pdf). We find a lifetime of 0.13 ns for all fits. We stress however that this estimate is to be handled with extreme care. Our measurements are carried out at 2.5 K, which fixes the resolution of the dI/dV spectra to 1 meV (the resolution is $5.5 k_B T$, where k_B is the Boltzmann constant). This corresponds to the width of the inelastic feature in our spectra as can be seen for the derivatives of the dI/dV (Fig. 1b). Hence, our feeling is that we cannot properly estimate the lifetime effects in this way contrary to the work of Loth et al. where the temperature was of 0.5 K. This might also explain why our fits are insensitive to the value of z . Given our experimental setup, we therefore prefer not to address lifetime effects. This will definitely be appealing for future work, for example using a pump-probe setup.

Comment 7:

There is a recent paper by Jacobson et al. (Science Advances 3, e1602060 (2017)) which also report on switching the spin state from $S=1$ to $S=1/2$ by the amount of hydrogen atoms on a Co atom. Even though that the switching is due to a different effect, the authors should consider citing this work in their introduction.

Author Reply 7:

We have added the reference of this nice paper.

Comments of Reviewer 2

Reviewer 2 finds that the study presents an important advance in the field of spintronics and nanomagnetism, with well-supported conclusions. He/She recommends the publication of the paper in Nature Communications after considering the following minor points:

Comment 1:

In Fig. 2(a) there is no colour coding of the 2D intensity plot.

Author Reply 1:

As suggested by the referee, we have included a colour coding for the 2D intensity plot.

Comment 2:

In the main manuscript is not clear how the d_{xz} and d_{yz} orbitals are oriented with respect the tip-substrate geometry.

Author Reply 2:

The xy plane is parallel to the Cu(100) surface (x aligns to $\langle 100 \rangle$ the direction) and z is the transport direction. Frontier orbitals shows C(2p) and Ni (3dxz-3dyz) contributions. The following figure shows the doubly-degenerated frontier molecular orbitals for exactly the same geometry as the molecule has at $d=9.66\text{\AA}$ (the isovalue was increased to see only the Ni (dxz-dyz) contribution).

Following the indication of the referee, we have included this picture in the Supplementary Note 4 as well as a reference to it in the main text. In Pag. 6, now it reads:

The transmission of Fig. 4 (a) implies that the transport is mainly due to the hybridization of surface electronic states with the frontier molecular orbitals, which are dxz – and dyz –based molecular orbitals (see Supplementary Note 4) [14].

Comment 3:

There are spelling mistakes in the manuscript that needs to be corrected.

Author Reply 3:

We have checked the spelling mistakes to our best.

REVIEWERS' COMMENTS:

Reviewer #1 (Remarks to the Author):

I am very happy to see that the authors have addressed mine and the second referee's questions and comments.

I can now fully recommend publication of the manuscript.